# VirClust—A Tool for Hierarchical Clustering, Core Protein Detection and Annotation of (*Prokaryotic*) Viruses

**DOI:** 10.3390/v15041007

**Published:** 2023-04-19

**Authors:** Cristina Moraru

**Affiliations:** Institute for Chemistry and Biology of the Marine Environment, Carl-von-Ossietzky–Str. 9-11, 26111 Oldenburg, Germany; liliana.cristina.moraru@uni-oldenburg.de

**Keywords:** VirClust, virus genome clustering, virus protein clustering, virus protein annotation, virus classification, phage classification, core proteins, shared viral proteins, virus protein clustering

## Abstract

Recent years have seen major changes in the classification criteria and taxonomy of viruses. The current classification scheme, also called “megataxonomy of viruses”, recognizes six different viral realms, defined based on the presence of viral hallmark genes (VHGs). Within the realms, viruses are classified into hierarchical taxons, ideally defined by the phylogeny of their shared genes. To enable the detection of shared genes, viruses have first to be clustered, and there is currently a need for tools to assist with virus clustering and classification. Here, VirClust is presented. It is a novel, reference-free tool capable of performing: (i) protein clustering, based on BLASTp and Hidden Markov Models (HMMs) similarities; (ii) hierarchical clustering of viruses based on intergenomic distances calculated from their shared protein content; (iii) identification of core proteins and (iv) annotation of viral proteins. VirClust has flexible parameters both for protein clustering and for splitting the viral genome tree into smaller genome clusters, corresponding to different taxonomic levels. Benchmarking on a phage dataset showed that the genome trees produced by VirClust match the current ICTV classification at family, sub-family and genus levels. VirClust is freely available, as a web-service and stand-alone tool.

## 1. Introduction

Viral classification and taxonomy have recently undergone major changes. The Baltimore classification scheme, based solely on the viral nucleic acid type has been replaced by a viral megataxonomy, based on viral genome features, including shared genes (proteins) [1]. The traditional five-rank structure of viral taxonomy was replaced by a fifteen-rank classification hierarchy, similar to the Linnaean taxonomy [2]. As a catalyst for these changes, the unparalleled insights into virus genome organization and evolution were facilitated by the advent of genome sequencing.

The first of the four principles for viral taxonomy recently established states that “virus taxonomy should reflect the evolutionary history of viruses” [3]. Traditionally, reconstruction of the evolutionary history is achieved through phylogenetic analysis of conserved genes. The best-known example is that of the rRNA or ribosomal genes, which are both universally present and conserved in cellular organisms. In contrast, viruses share no universal gene and likely have multiple points of origin [4,5,6,7]. Therefore, traditional phylogenetic methods, in which phylogenetic trees are constructed based on multiple alignments of homologous genes (proteins) universally present in all viruses, cannot be applied to viruses as a whole.

Gene (protein) sharing networks have been used to explore how viruses are related to each other [8] and resulted in the definition of viral hallmark genes (VHGs), which represent genes broadly found in diverse virus groups, but are not universally present. Based on the presence of such VHGs, six viral realms have been defined to date: *Adnaviria*, *Riboviria*, *Monodnaviria*, *Duplodnaviria*, *Varidnaviria* [1,9], and *Ribozyviria* [10]. Prokaryotic viruses, infecting bacteria and archaea are spread through these realms, with the known majority belonging to class *Caudoviricetes* (former *Caudovirales* order) within *Duplodnaviria*. Many other viruses, cultivated and uncultivated, are yet unassigned to any realm, awaiting further evidence to classify them into an already existing realm or to a brand new one [1].

Inside the realms, viruses are further organized into hierarchical taxons, from kingdom to species, similar to the cellular world [2]. The methodologies for virus classification vary with the taxonomic rank and they have been recently proposed as a community-wide consensus [3]. At lower-level ranks — species and genus — the classification should be based on genetic relationships as calculated from multiple alignments of complete genomes or genes, supported by clustering methods based on intergenomic nucleic acid identities, reported for example by VIRIDIC [11]. At intermediary-level ranks—family, order and class — phylogenetic analysis of VHGs specific for that particular virus group should be used, combined with the analysis of the gene content and genome organization. Further on, at the highest-level ranks—phylum, kingdom, and realm — viruses should be classified based on specific and highly conserved VHGs and protein structure analysis.

Currently, there are several whole proteome-based virus classification tools, which have been used for the delineation of intermediary ranks. They can be classified into tools based on (i) whole proteome similarity, for example ViPTree [12] and VICTOR [13], (ii) on protein profile hidden Markov models (PPHMMs) and genomic organization models (GOMs), as implemented in GRAViTy [14,15], and (iii) on shared protein clusters, as implemented in vConTACT [16,17]. VICTOR and VipTree calculate pairwise intergenomic distances based on protein-protein BLAST comparisons of the whole viral proteomes in a given dataset and use them for hierarchical clustering of the respective viruses. GRAViTy uses concatenated proteins of the query viruses to search against pre-calculated databases of viral PPHMMs and GOMs and then computes for each query virus a PPHMM and GOM signature. These signatures contain information about the degree of similarity between the query and the databases and are used to calculate intergenomic pairwise distances, followed by hierarchical clustering of the viruses. Finally, vConTACT computes for the given dataset of viral genomes (including or not including a reference database) all protein clusters, based on BLASTP comparisons. Then, it uses the absence/presence of protein clusters to calculate intergenomic similarities between viruses, which are further used to construct a viral genome monopartite network. This method produces single-level viral clusters, potentially of the genus or family rank. The main disadvantages of these tools are that they either do not identify the genomic features (proteins) contributing to the clustering of the viruses (ViPTree, VICTOR, GRAViTy), or they do not produce hierarchical clusters (vConTACT).

The new taxonomic principles place core proteins and VHGs at the center of virus classification [3]. Their identification requires most often sensitive methods of recognition of protein homology (e.g., at HMM level) and calculation of the core proteins. With the idea of developing a tool that combines virus clustering into groups of different taxonomic levels with the detection and annotation of core proteins and of VHGs by sensitive and flexible methods for protein homology, I have created VirClust (Virus Clusterer). VirClust enables viral classification, by bringing to the table the following: (i) calculation of protein clusters, using highly sensitive methods for homology detection (BLASTP, followed by HMM comparisons); (ii) calculation of intergenomic distances based on the presence/absence of protein clusters; (iii) hierarchical clustering of the viral genomes based on their respective intergenomic distances; (iv) calculation of core protein clusters; and (v) protein annotation based on a state-of-the art collection of sequence databases. VirClust is available both as a web-service (virclust.icbm.de) and as a stand-alone command-line tool.

## 2. Materials and Methods

### 2.1. VirClust—Development and Workflow

VirClust was developed in the R v4.2 [18] programming language (https://cran.r-project.org/bin/windows/base/old/4.2.0/, accessed on 5 May 2022). The web interface was developed under the Shiny web application framework (https://cran.r-project.org/web/packages/shiny/index.html, accessed on 5 May 2022, RStudio, Boston, MA, USA). The stand-alone tool for Linux was wrapped in a container using the Singularity v. 3.5.2 software (https://sylabs.io/, accessed on 23 April 2020, Sylabs.io, San Francisco Bay Area, CA, USA). The stand-alone version can be deployed on any system running the Singularity software.

A complete VirClust workflow (Figure 1) is organized into three branches: (i) Branch A, based on protein clusters, (ii) Branch B, based on protein superclusters, and, (iii) Branch C, based on protein super-super clusters. Each branch is organized into four modules: (i) protein clustering; (ii) genome clustering; (iii) calculations of core proteins; and (iv) protein annotations. Each module consists of one or several steps.

For each given input, VirClust defines “projects”, which have a corresponding folder and an ID (the name of the folder). This folder is not directly exposed to the user in the web version, but it can be indirectly accessed through the project ID and the download buttons. In this folder, VirClust will save all the results, intermediary files, and status reports corresponding to the respective project. Within a project, all three branches or only some of them can be run, each branch either partially or completely. The three branches depend on each other at the level of the protein clustering modules. Branch B requires the protein clustering step from Branch A, and Branch C requires the protein clustering step from Branch B. Within a branch, most of the steps have a linear dependency amongst each other, meaning that they can be performed only if the previous steps have been already performed. In the stand-alone version, the prerequisite steps are automatically activated. For example, if the user chooses to run step 4A from Branch A, the prerequisite steps will be automatically activated and performed. In the web-server version, however, the user has to run the steps sequentially, one by one, and the corresponding elements from the graphical interface only become active after the prerequisite steps have been calculated. Within a project, both in the stand-alone and the web-server versions, the user can choose to recalculate certain steps (e.g., using different parameters). In this case, all the results from the steps depending on the re-calculated step will be automatically removed from the project. For example, if the user chooses to re-calculate step 2A, which performs BLASTP-based protein clustering, the results from all the next steps in Branch A, as well as from all steps from Branch B and C, will be deleted. A scheme defining the dependencies between the steps is given in Figure 1.

Each step produces several files, of which some are of interest to the user and are referred to from here on as “usable outputs”. These files can be retrieved by the user either by downloading from the webpage (when using the VirClust web-service) or directly from the disk space when using the VirClust stand-alone version.

Several operations are computationally intensive and some of them, e.g., BLASTP (step 2A) and bootstrapping (steps 3A, 2B, and 2C), have been parallelized. The computational time can increase significantly with the number of proteins/P(SS)Cs, especially during bootstrapping.

#### 2.1.1. Protein Clustering Module from Branch A

This module consists of two steps, one for gene prediction and translation, and the other for protein clustering based on BLASTP.

##### Gene Prediction and Protein Prediction—Step 1A

In the first step, VirClust uses MetaGeneAnnotator [19] to predict genes in each viral genome, and then the seqinr R package [20] to translate the predicted genes. The user can choose the genetic code for translation, the default being the one for bacteria and archaea (11). The usable outputs from this step are: (i) the protein files, consisting of a single file (.faa format) containing the proteins from all genomes, and a folder with a protein file (.faa format) per genome; and, (ii) a table in .tsv format containing all the predicted genes (including start, end, length, etc) and their corresponding proteins for every viral genome. In this step, each protein receives and is saved with a unique identifier (protein ID) that does not include the genome name, to prevent possible problems in the upstream steps due to varying genome name formats and lengths. The correspondence between the protein and its corresponding gene and genome can be retrieved from the genome and protein table.

##### From Proteins to Protein Clusters—Step 2A

In the second step, VirClust groups similar proteins into protein clusters (PCs). First, it compares all proteins with each other using BLASTP from the BLAST+ package [21]. The BLASTP hits (query-subject pairs) are filtered based on their e-value, bitscore, coverage (of both subject and query), and identity. By default, hits are kept if bitscore > 50, e-value < 0.00001, coverage > 0, and identity > 0. Further, the remaining hits are used to cluster the proteins based on their (i) e-values, (ii) log10 transformed e-values, capped at 200 (the default) (iii) bitscore, or (iv) normalized bitscores (maximum from “bitscore for prot1-prot2 hit/bitscore for prot1-prot1 hit” and “bitscore for prot2-prot1 hit/bitscore for prot2-prot2 hit”). The clustering is performed by mcl (https://micans.org/mcl/ (accessed on 27 February 2018)), with the options “-I 2 -abc -o”. The usable output from this step is the genome-protein table from step 1A, to which a column with the corresponding PCs for each protein has been added.

#### 2.1.2. Protein Clustering Modules from Branch B and Branch C

The protein clustering steps from Branch B and Branch C are similar, in the sense that they are both based on Hidden Markov Model (HMM) similarity and thus, they use almost the same options and algorithms. In both branches, this module has only one step—1B (Branch B) or 1C (Branch C).

##### From Protein Clusters to Protein Superclusters—Step 1B

In step 1B, VirClust groups the PCs and their corresponding proteins into protein superclusters (PSCs), based on HMM similarities. First, for each PC calculated above it creates a multiple alignment with Clustal Omega [22], options “-pileup -iter = 2”. Then, it calculates hidden Markov models (HMMs) with hhmake (hhsuite package [23], options “-id 100 -diff 1,000,000”). Further, it compares all HMMs with each other using hhsearch (hhsuite package, options “-id 100 -diff 0 -p 50 -z 1 -Z”). The results of this comparison are filtered based on probability, coverage, and alignment length, with thresholds established by the user. The default thresholds for keeping the results are those previously used for organizing dsDNA viral genomes into a bipartite network [8]: probability ≥ 90, subject coverage ≥ 50 and then, probability ≥ 99, subject coverage ≥ 20, alignment length ≥ 100. Finally, the hits passing the thresholds are used to cluster the PCs into PSCs, using mcl (options “ -I 2 -te 20 -o”). Similar to step 2A, the clustering of the HMM hits can be done based on their (i) e-values; (ii) log10 transformed e-values (default); (iii) score; and (iv) normalized score. The usable outputs from this step are: (i) a .zip archive with the multiple alignment for each PC, in aligned multifasta format; and (ii) the genome-protein table from step 2A, to which a column with the corresponding PSCs for each protein has been added.

##### From Protein Superclusters to Protein Super-Superclusters—Step 1C

In step 1C, VirClust groups PSCs into protein super-super clusters (PSSCs). For this, after creating a protein multiple alignment for each PSC, it proceeds similarly to step 1B. The exception is the processing of the multiple alignment, which now includes the removal of columns if they are made up of more than 50% gaps. The usable outputs from this step are: (i) multiple alignments corresponding to each PSC, in aligned multifasta format; and (ii) the genome-protein table from step 1B, to which a column with the corresponding PSSCs for each protein has been added.

From here on, the term P(SS)C is going to be used when referring generally to clusters of proteins, instead of using the wordier phrase “PC, PSC, or PSSC”.

#### 2.1.3. Genome Clustering Modules from Branches A, B, and C

The genome clustering modules in the three branches are similar. They each have two steps: (i) hierarchical clustering of viral genomes (step 3A in Branch A, 2B in Branch B, and 2C in Branch C); and (ii) splitting of the hierarchical clustering tree into smaller viral genome clusters (VGCs) and calculation of the corresponding statistics (step 4A in Branch A, 3B in Branch B and 3C in Branch C). The difference between the three branches lies in the input received by the first step in the module. In Branch A, step 3A takes as input the PCs generated in step 2A. In Branch B, step 2B takes as input the PSCs generated at step 1B. In Branch C, step 2C takes as input the PSSCs generated at step 1C. Each of the two steps also has a data visualization sub-step, which enables a quick graphical representation of the results.

##### Hierarchical Clustering of the Viral Genomes Based on Their P(SS)C Content—Steps 3A, 2B or 2C

VirClust first calculates pairwise intergenomic distances, based on the P(SS)C content of each viral genome. For this, the presence of a P(SS)C in a viral genome is rewarded a score of 1, irrespectively of how many P(SS)C replicates are found in the genome, and the absence of a P(SS)C is rewarded a score of 0. Pairwise distances are calculated using the formula:DistAB = 1 − (2 × PssCsAB)/((PssCsA + PssCsB))(1)
where,

PssCsAB = score sum for all P(SS)Cs in common between genome A and genome B

PssCsA = score sum for all P(SS)Cs present in genome A

PssCsB = score sum for all P(SS)Cs present in genome B

Further, VirClust performs a hierarchical clustering of the viral genomes based on the above-described intergenomic distances. For clustering, it uses either the stats 3.5 package, without bootstrapping (default), or the pvclust 2.2 package [24,25], when bootstrap resampling is desired. The “complete” agglomeration method is used as default, with the other option being “average”. Following bootstrap resampling, the pvclust package calculates and reports three probability values for each cluster: (i) selective inference *p*-value (SI); (ii) approximately unbiased *p*-value (AU), and (iii) bootstrap probability (BP) value [25]. Due to the high CPU demand, the boot-strap option is inactivated if more than 50 genomes are inputted in the VirClust web-service, but is fully available in the stand-alone version. Usable outputs from this step are: (i) a matrix-like table in the .tsv format, containing the calculated intergenomic distances; and (ii) a tree file in the .newick format, containing the clustering results (the hierarchical tree). If bootstrapping is performed, then three .newick files are generated, one each for the SI, AU, or BP values. In addition to these outputs, the data visualization sub-step from step 3A (or 2B/2C) allows the user to generate and download a PDF file containing an ordered and color-coded heatmap of the intergenomic similarities (calculated as “(1 − intergenomic distance) × 100”).

##### Splitting into Viral Genome Clusters and Related Statistics—Steps 4A, 3B or 3C

In this step, VirClust can split the viral genomes into clusters, by “cutting” the tree previously calculated (in step 3A for Branch A, 2B for Branch B, and 2C for Branch C) at user-defined distances. The default distance is 0.9. This tree cutting is performed with the “stats” package from R. The resulting viral genome clusters (VGCs) will contain viral genomes that are more similar to each other than to other genomes, depending on the intergenomic distance used for tree splitting.

Then, for each viral genome, VirClust calculates the following statistics: (i) total number of proteins in the genome (corresponds to the total gene number); (ii) total number of proteins that belong to singletons (P(SS)Cs containing only one protein, that is P(SS)Cs which are not shared with any other virus in the dataset); (iii) total number of proteins found in P(SS)Cs shared with other viral genomes in the dataset; (iv) total number of proteins found in P(SS)Cs shared with viral genomes from the same VGC, regardless if they are shared with viral genomes from outside the VGC as well; (v) total number of proteins found in P(SS)Cs shared exclusively with viral genomes only from the same VGC; (vi) total number of proteins found in P(SS)Cs shared with viral genomes from other VGCs, regardless if they are shared with viral genomes from the same VGC as well; (vii) total number of proteins found in P(SS)Cs shared exclusively with viral genomes from other VGCs; (viii) Silhouette width (calculated with the R package “cluster” [26]). A table (.tsv format) with all these statistics is available to the user as usable output.

Finally, a table is prepared in which the rows represent the viral genomes, ordered similarly to the branches in the tree, and the columns represent the shared P(SS)Cs. The column order is based on their clustering with the “stats” R package, using the “binary” distance and the “complete” agglomeration method. This table is used for further steps, and it is also available as a usable output, as a .tsv file.

In addition to the two tables prepared for the complete data set, VirClust returns for each VGC: (i) a folder with a .faa file for each genome in the VGC; (ii) a .fna file with all genomes in the VGC; (iii) a table with genome statistics only for the genomes in the respective VGC; and (iv) a table with the pairwise intergenomic similarities for the genomes from the respective VGC.

The data generated in this step (4A/3B/3C), together with the tree calculated in the previous step (3A/2B/2C), can be used to generate an integrated figure in the data-visualization sub-step. Here, VirClust uses the R package ComplexHeatmap v. 2.5.3 [27] to generate a visual representation of the genome clustering. This is composed of: (i) the genome clustering tree; (ii) a heatmap documenting the presence/absence of the different P(SS)Cs in the viral genomes; (iii) several annotations documenting the genome and protein statistics, the Silhouette width and the cluster designation. If the genome tree has been split into several clusters, the heatmap and the corresponding annotation are split as well. The usable output from this step is a file in .PDF format.

#### 2.1.4. Core Proteins Modules from Branches A, B, and C

The core protein modules from the three branches are similar and each has only one step. Their inputs differ: step 5A (Branch A) takes PCs as input, step 4B (Branch B) takes PSCs as input and step 4C (Branch C) takes PSSCs as input. The core proteins represent P(SS)Cs found in all genomes from a VGC. VirClust calculates the core proteins for each VGC generated in the genome clustering module from the respective branch. For each VGC, the following usable outputs are generated: (i) a table (.tsv and .RDS format) containing for each genome in the VGC its core proteins, their assignment to P(SS)Cs, their corresponding genes, and their features (genome location, length, etc.); and (ii) two .faa files, each containing all core proteins for the respective VGC, labeled in one case with the VirClust protein ID and, in the other case, with a name composed of their P(SS)C number, genome name, and gene number.

#### 2.1.5. Protein Annotation Modules from Branches A, B, and C

The protein annotation modules from the three branches are similar. They each take as input either all the proteins in the dataset or only the core proteins calculated in the respective branch. VirClust annotates each protein by comparing it with several databases. For each database, this process takes place in two phases. In the first phase, homologs are searched for all proteins, and only the best matches (see below) are kept for each of them. In a second phase, these annotations are added to the genome-protein table from steps 2A/1B/1C, depending on the branch. These tables are identical, except for the columns containing protein clustering information: in Branch A the table contains the column for PCs assignment, in Branch B it contains the columns for PCs and PSCs assignment, and in Branch C it contains the columns for PCs, PSCs, and PSSCs assignments. After all desired databases have been queried, all the annotation results can be merged into a single table. These tables represent the usable outputs and can be downloaded as .tsv files.

The NR database from NCBI is searched using BLASTP (“-evalue 0.0001 –max_target_seqs 1000”). From the results, hits are removed if they represent hypothetical proteins, have a query/subject coverage < 40, have a pident < 30, or a bitscore < 50. From the remaining hits, that with the higher bitscore is used to annotate the query protein.

The prokaryotic Virus Orthologous Groups (pVOGs) database [28], the Virus Orthologous Group database (VOGDB, https://vogdb.csb.univie.ac.at (accessed on 2 August 2021), [29]), and the Prokaryotic Virus Remote Homologous Groups (PHROG) database [30] are searched using hhsearch [31] (“-id 100 -diff 0 -p 50 -z 1 -Z 600”). Only hits with an e-value lower than 0.01 are kept. For each database, the hit with the highest score is used to annotate the query protein.

The efam and efam-XC databases are searched using hmmscan [32], options “-E 0.01”, followed by result removal if score < 40. For each database, the hit with the highest score is used to annotate the query protein.

The InterPro database [33] is searched using InterProScan [34]. Results with the description “Domain of unknown function” and IP analysis “MobiDBLite” are removed.

### 2.2. Running VirClust on Test Datasets

For testing VirClust, a first dataset of 1951 genomes (see Appendix A) was selected from the dsDNA bacterial viruses currently recognized by the International Committee on Taxonomy of Viruses (ICTV). The dataset, here named dsDNA_DB, included viruses from two viral realms: *Duplodnaviria* and *Varidnaviria*. From each genus, a maximum of four representatives were selected. In total, this dataset contained viruses organized in 5 orders, 40 families, 103 subfamilies, and 1144 genera.

A second dataset of 887 phages, named here Fam_DB, was built from the dsDNA_DB, by keeping only phages having a family affiliation. This resulted in a total of 40 families, 68 subfamilies, and 545 genera.

The genomes in the dsDNA_DB and Fam_DB datasets were clustered based on PCs and PSCs, using the default VirClust parameters. The trees generated in steps 3A and 2B were imported in iTol [35] and annotated with taxonomic information. Additionally, the trees from Fam_DB were split into clusters using various distances, from 0.05 to 0.99. Then, for each VGCs and each family, subfamily, and genus, the following were calculated: (i) how many of the respective taxons were present per VGC, and (ii) how many VGCs were present per taxon.

A third, smaller dataset, containing only 37 phages and named here Crz_DB, was built by keeping from the Fam_Db only members of *Chaseviridae*, *Rountreeviridae*, and *Zobellviridae*. The Crz_DB was used for further illustration of the different VirClust features.

## 3. Results and Discussion

### 3.1. VirClust—A Tool for Viral Genome Clustering, Core Protein Detection, and Protein Annotation

VirClust is a multifaceted viral genome analysis tool, developed to assist in the clustering of prokaryotic viruses, including for taxonomic classification, and functional annotation of their protein-encoding genes. To enable viral classification, on the one hand side it performs a hierarchical clustering of the viral genomes, which can be used to group viruses at different taxonomic levels, and on the other hand side it identifies core proteins, which can be used for further phylogenetic analysis. To enable protein annotation, VirClust searches for homologous proteins within seven different protein sequence and HMM profiles databases.

VirClust is organized into three branches, based on PCs, PSCs, and PSSCs, respectively. Each branch is organized into four modules (see Figure 1): (i) protein clustering; (ii) genome clustering; (iii) calculation of core proteins and (iv) protein annotation.

In the first module, VirClust performs a series of basic steps (see Figure 1): (i) protein prediction and translation (ii) protein grouping into PCs, based on BLASTP detectable homologies (Branch A); (iii) PC grouping into PSCs, based on HMM profile search detectable homologies (Branch B); and (iv) PSC grouping into PSSCs, again based on HMM profiles (Branch C). HMM profiles have been successfully used in previous studies to group viral protein clusters into superclusters [8,30] because they capture more distant relationships between proteins. Grouping of PSCs into PSSCs, however, is not a widely spread methodology. It was implemented in VirClust to allow the community to explore finding even more distantly related protein and it was already used successfully for the clustering of ssDNA phage proteins and genomes [36]. The most important clustering parameters have been exposed to the user and are adjustable, to enable protein clustering for a wide range of applications.

In the genome clustering module, VirClust uses the presence/absence of P(SS)Cs in viral genomes to calculate intergenomic distances. These can be transformed into intergenomic similarities by the formula “(1 − distance) × 100” (e.g., later, in step3A_Plot, step2B_Plot, and step2C_Plot), which reflect the proportion of shared P(SS)Cs between two genomes.

These distances are further used to cluster viruses hierarchically and then to split them into VGCs based on a user-defined distance threshold. Depending on the distance threshold used, the VGCs can contain more closely or distantly related viral genomes. Therefore, VGC calculated for different distance thresholds potentially represent different taxonomic ranks.

To evaluate the genome clustering, several indicators are calculated. The shared protein statistics and the Silhouette width are genome-specific statistics that can be used to appraise the affiliation of individual viruses to VGCs. The proportion of proteins shared with any other viral genomes in the analyzed dataset shows what proportion of all proteins from a single virus is used for clustering. If only a small proportion of proteins are shared with other viruses in the dataset, it can increase the clustering uncertainty, because the singletons can hide relationships with yet unknown viruses and potentially, a different clustering. The Silhouette width measures, on a scale of −1 to 1, how related a virus is with other viruses in the same VGCs. Values closer to 1 indicate higher similarity to members of its own VGC. Values closer to −1 indicate higher similarity with viruses in other VGCs. Similar to a negative Silhouette width, a high proportion of proteins shared outside its own VGC can indicate an incorrect clustering. In addition, if the bootstrapping option is chosen, for each cluster in the hierarchical tree, three different probability values (SI, AU, and BP) can be calculated by bootstrapping the P(SS)Cs and can be used to assess the clustering uncertainty [24,25].

Two data visualization sub-steps are built-in in the genome clustering module. The first one outputs a clustered heatmap of the intergenomic similarities between all genome pairs (step3A_Plot, step2B_Plot, and step2C_Plot, see Figure 1 and Figure 2). The second one outputs an integrated visualization (step4A_Plot, step3B_Plot, and step3C_Plot, see Figure 1 and Figure 3) of the hierarchical clustering of viruses, of the distribution of their protein content and their grouping in VGCs, with the corresponding statistics. The protein content of the viral genomes is visualized as a heatmap, in which the columns represent P(SS)Cs and the rows represent viral genomes, ordered according to the hierarchical clustering tree. Only shared proteins are depicted in the heatmap, the proportion of singletons being shown as an annotation along the heatmap (see Figure 3, “shared proteins” statistics). Together, the heatmap and the annotated statistics allow the opening of the “black box” of the tree. The user can visualize and thus identify which P(SS)Cs have contributed to the hierarchical clustering, can identify which distance threshold is best for splitting the tree into VGCs, and also, can judge the quality of the clustering. Furthermore, the heatmap allows the identification of P(SS)Cs characteristic for certain viral groups and of potential gene duplication/gene split events (by the increased number of a P(SS)C in a viral genome).

In the third module, VirClust calculates the corresponding core proteins for each of the VGCs identified at steps 4A, 3B, or 3C. The core proteins are defined as those P(SS)Cs present in all the genomes from a VGC, regardless of their copy number per genome. The suitability of each identified core P(SS)C to be further used for phylogenetic analyses should be judged by the user from their multiple alignments (provided for download) and from their functional annotations (provided by the fourth module). P(SS)C subjected to gene duplication events, which can lead to truncated proteins, as well as those having gene insertions (for example homing endonucleases, commonly spread in polymerases) should be carefully evaluated. Furthermore, proteins composed of multiple domains (for example DNA polymerases) can be encoded by a single gene or by more genes, each for a single domain. Depending on the parameters from the protein clustering steps, the genes for the multiple domains can be grouped in a single P(SS)C (more in the section “Protein clustering—parameters choice”). The use of these P(SS)Cs for phylogenetic analysis should be carefully evaluated and eventually, the single domain proteins concatenated.

In the fourth module, VirClust annotates selected proteins (either all proteins or only the core P(SS)Cs) by comparing them with several sequence and HMM profiles databases (see Figure 1). Each database can be queried separately. The best hits from each database are identified for each protein. The results are then integrated into a single table, together with the information about the genome localization of each protein, and the assignment to P(SS)Cs and VGCs. The final annotation, integrating the information from all queried databases should be decided by the user, during the careful evaluation of the annotation table. The protein assignment to P(SS)Cs greatly facilitates the annotation of those proteins without significant hits with any databases, because proteins grouped in the same P(SS)C should in general have the same function. The exceptions are those proteins composed of multiple domains or those with insertions. The evaluation of the annotation results and the multiple alignments enables the identification of multiple domains, as well as potential gene insertions.

### 3.2. Availability

The VirClust web-service (virclust.icbm.de) provides a graphical interface for running VirClust remotely. To avoid a heavy burden on the hosting server, it should be used only for small and medium-sized projects. Larger projects can be analyzed with VirClust stand-alone, which can be downloaded from virclust.icbm.de and installed on the user’s servers, run from the command line, and integrated into bioinformatics pipelines. Both VirClust web and stand-alone come with complete and comprehensive user manuals, available at virclust.icbm.de and also here, as supporting information (see SI Files S2 and S3).

### 3.3. Protein Clustering—Parameters Choice

The protein clustering into PCs, and further into PSCs and PSSCs, represents the foundation on which the clustering of the viral genomes is based. This is a two-step process, in which first homologs are detected based either on BLASTp (for PCs) or HMM searches (for PsSC, PSSC), and then the proteins are clustered based on the found homologies. Therefore, the parameters for defining homologs will influence which proteins cluster together, which in turn will influence which viruses cluster together in the genome clustering module.

In the first step, the filtering of the search results is critical for the placement of proteins into P(SS)Cs. In the case of PCs, VirClust filters the search results based on their e-value, bitscore, alignment coverage, and percent identity of the two sequences. Different studies have been used for protein clustering different combinations of these parameters; for example, only e-value and bitscore [37], e-value and coverage [38], or different values of the parameters, or even, no filtering at all [39]. In the case of PSCs and PSSCs, VirClust filters the search results based on their probability, coverage, and HMM alignment length, similar to Iranzo et al. [8]. Here, to enable the detection of more distant homologs, the authors performed a two-tier filtering, selecting all hits with (i) a probability higher than 90% and coverage >50% and (ii) a probability higher than 99%, but a coverage >20% and a minimum length of 100. In the second step, VirClust clusters proteins based on e-values, as in Roux et al. 2015, log-transformed e-values, as in Enright et al. 2002, normalized bitscore, as in Chan et al. 2013 [40] or bitscores.

VirClust enables the use of all the above parameters for protein clustering, including the two-tier filtering step for HMM results. However, rather than imposing strict values for these parameters, VirClust allows the user to set her/his values, in addition to the default suggestions.

To illustrate the influence of the search results filtering step on the protein and genome clustering, the Crz_DB dataset was subjected to protein clustering with three different sets of parameters, and then intergenomic distances and genome trees were calculated with the default parameters. Initially, the coverage and percent identity were set to 100%, to group into PCs only identical proteins. Then, the coverage and identity were progressively decreased to the VirClust default parameters, to allow for finding more distant protein homologs (see Table 1). With relaxing the filtering parameters, the number of PCs formed by the two thousand and twelve proteins of the Crz_DB dataset decreased from one thousand nine hundred and eighty-two PCs when only identical proteins were allowed to group, to eight hundred and five PCs when the default parameters were used (see Table 1). The more homologous proteins were found by relaxing the filtering parameters, the more PCs the viral genomes had in common (see heatmaps in Appendix A) and thus, the lower the intergenomic distances between them (see Table 1 and Appendix A). Because with the first, stringent set of parameters, the intergenomic distances for most genome pairs equaled one (no PCs in common between the respective genome pairs), most viral genomes did not group into clusters in the genome tree (see Appendix A). Decreasing the intergenomic distances resulted in more viral genomes forming clusters (see Appendix A), such that, with the default VirClust parameters three large genome clusters were obtained, corresponding to the three families in the dataset—*Chaseviridae*, *Rountreeviridae*, and *Zobellviridae* (see Appendix A).

One other important aspect is the clustering of proteins composed from multiple domains, which in some viruses can be encoded by the same gene, and in others separately. This is the case of the DNA polymerases from *Zobellviridae*, and we used this dataset to determine how the different filtering parameters can influence the clustering of such proteins. Most genomes in this family have a DNA polymerase gene with an exonuclease and a polymerase domain. However, a few of them have the two domains as independent genes. From the three hit filtering parameters, the coverage parameter will most likely influence how these proteins will be clustered. When clustered with the default settings (for PCs—coverage = 0%; for PSCs—coverage 1 = 50% and coverage 2 = 20%), in which the coverage does not play a significant role, all DNA-pol related proteins (having both domains, or just the exonuclease domain, or just the polymerase domain) clustered in a single PSC. When increasing the coverage threshold (for PCs: coverage = 70%; for PSCs—coverage1 and coverage 2 = 60%), the exonuclease and the polymerase domains clustered in different PSCs. However, also the DNA polymerase domains were split into three PSCs, indicating that recognition of more distantly related homologs was hampered by the increased coverage threshold (see Appendix A), even when using the more sensitive HMM searches. Because multidomain proteins represent usually a small proportion of all viral proteins, their lumping or splitting into clusters most often will not influence significantly the intergenomic distances, and thus, the clustering of the viral genomes. However, in the case of protein annotation, lumping can mislead the user to believe that the two domains of the protein have the same function. Therefore, protein clustering can be performed with a different set of parameters for protein annotation than for genome clustering. Because the protein IDs are always the same, the two protein clustering data sets can be reconciled afterward (e.g., through table join operations).

Due to the flexibility of the protein clustering parameters, VirClust can be used to cluster viral genomes for very different purposes. For example, if the purpose is to discriminate between highly related viral genomes, and to detect even small sequence variations between proteins, then proteins should form clusters only if they are identical. On the other hand, if the purpose is to detect more distant relationships between viral genomes and to place them for example into family-level taxons, then parameters allowing the clustering of more distantly related proteins should be used.

### 3.4. VirClust Hierarchical Clustering Matches ICTV Virus Classification

Two datasets, dsDNA_DB and Fam_DB, the second representing a subset of the first, were used to test the ability of VirClust’s default protein and genome clustering parameters to capture relationships between viruses at different taxonomic levels. Both datasets contained phages from *Duplodnaviria* and *Varidnaviria*. Within these two realms, as per the official ICTV classification at the time of data analysis, all phages were classified within genera or high-ranking taxons (realm, kingdom, phylum, and class). However, only part of them was classified into middle-ranking taxons, e.g., family and order. As a result, from the one thousand nine hundred and fifty-one phages in the dsDNA_DB, only eighty-three phages were assigned to orders and eight hundred eighty-two phages were assigned to families. This motivated the use for part of the analyses of Fam_DB, which comprises the subset of phages in the dsDNA_DB having a family classification.

First, the genomes in dsDNA_DB were clustered based on PCs and PSCs, and the resulting genome trees were compared with the current ICTV taxonomy (see SI Files S6 and S7). Within these trees, the viral genomes formed several major clusters, branching almost independently from each other, due to the large intergenomic distances between them. The high-ranking taxons, that is realm, kingdom, phylum, and class, did not form individual clusters, neither in the PC or PSC tree. For example, at the realm level, the *Varidnaviria* phages were split among seven major clusters (Appendix A). At the order level, there were differences between the PC and PSC-based trees. The two larger orders in the dataset, *Crassvirales*, and *Kalamavirales*, were split among two and three major clusters in the PC tree (Appendix A), but formed individual clusters in the PSC tree (Appendix A). From the family level down, phages belonging to the same taxon clustered together, both in the PC and PCS-based trees (Appendix A). The exception is the *Tectiviridae* family, which was split in the PC-based tree, but formed a single major cluster in the PSC tree.

Then, the genomes in Fam_DB were clustered, also based on PCs and PSCs. A PSC-based tree is shown in Figure 4, with the extended tree and its annotations being available in SI file 8. Here, the clustering together of phages from the same family is evident. Furthermore, quite often families are seen grouped into a larger cluster, potentially of the order level. This illustrates the potential of PSC-based genome clustering to capture also order-level relationships.

Taken together, these data show that PSC-based intergenomic distances, as calculated with the default protein clustering parameters, are unable to capture very distant relationships between viruses, as they take place at the class, phylum, kingdom, and realm levels. This is consistent with the known loss of protein sequence similarity at such large phylogenetic distances. However, these intergenomic distances will capture relationships at the order, family, subfamily, and genus level, and, at these levels, the hierarchical clustering produced by VirClust in general matches the current ICTV classification.

### 3.5. Distance Thresholds for Different Taxonomic Levels

Organizing viruses into hierarchical taxonomic ranks implies that some types of intergenomic distance thresholds can be assigned to each rank and used for classification purposes. Therefore, a hierarchical genome tree, such as that produced by VirClust, could be split into smaller clusters of different taxonomic levels, depending on the intergenomic distance used for the splitting of the tree (see examples of tree splitting in Appendix A). Such thresholds have been defined, for example, for the species and genus levels, by the Bacterial and Archaeal Viruses Committee of ICTV, but only for nucleic acid based clusters.

Here, to identify thresholds for different taxonomic levels, the PC and PSC-based trees from the Fam-DB dataset were first split with various intergenomic distances, from 0.05 to 0.99. Then, for each resulting VGC and each of the genus, subfamily, family, and order levels, the number of individual taxons per VGC was evaluated. Furthermore, for each taxonomic level, it was calculated how many VGCs were present per taxon, at different distances. These evaluations were performed separately on *Duplodnaviria* (Figure 5, Appendix A) and *Varidnaviria* (Figure 6, Appendix A). As expected, too low distance thresholds for a certain taxonomic level resulted in the splitting of taxons into multiple VGCs. Too high distance thresholds resulted in the clumping of several taxons within the same VGC (see Appendix A). Ideally, there would have been an intergenomic distance for each taxonomic rank, at which all VGCs corresponded to single taxons. In practice, however, because the phage taxons in the dataset were created using different methods, and thus, different distances, there was no single intergenomic distance at which no clumping or splitting of taxons occurred (Appendix A). The best threshold to use when creating new taxons is, therefore, that at which both clumping and splitting are minimal (Figure 5 and Figure 6). For *Duplodnaviria*, the following intergenomic distance thresholds can be recommended based on data evaluation: (i) for PC-based trees, 0.925 for family, 0.625 for subfamily, and 0.3 for genus levels; and (ii) for PSC-based trees, 0.9 for family, 0.6 for subfamily and 0.225 for genus. For *Varidinaviria*, the following can be recommended: (i) for PC-based trees, any distance between 0.6 and 0.725 for genus level; (ii) for PSC-based trees, 0.95 for order, 0.9 to 0.925 for family and 0.55 to 0.675 for genus level. For this realm, distance thresholds for the PC-based tree make sense only for the genus level, because the members of the main order and family do not form individual clusters.

It is evident from the data that across different virus realms different thresholds were used to delineate the same taxon ranks. For example, the genera in *Duplodnaviria* have a lower diversity and thus lower distance threshold, than the genera in *Varidnaviria*. It is up to the community to establish relevant thresholds, either unique for all realms, or different between realms. However, inside one realm, the thresholds should be coherent, to ensure that the viral diversity is hierarchically distributed and comparable across taxons.

### 3.6. Identification and Annotation of Core-Proteins

In the context of viral taxonomy and classification, core proteins refer to the proteins that are highly conserved across different members of a viral cluster (most often family-level cluster). These core proteins are typically involved in critical functions of the virus, such as replication, transcription, and packaging of the viral genome. They are often structural proteins that form the basic building blocks of the viral particle, or enzymes that are essential for the virus to replicate its genetic material. They are used to define the respective viral cluster and to infer phylogenetic relationships. In practical terms, to recognize the core proteins, one needs to group first the proteins into clusters of closer or more distant homologs. In VirClust, these clusters are represented by PCs, PSCs, or PSSCs. Therefore, in VirClust, core proteins are represented by those P(SS)Cs shared by all members of a VGC.

To illustrate the ability of VirClust to detect core proteins, the shared PSCs for each family-level VGC (intergenomic distance threshold of 0.9) in Fam_DB were calculated and further annotated against the PHROG database. At an intergenomic distance threshold of 0.9, Fam_DB was split into thirty-six VGCs. Most VGCs corresponded to one phage family, six VGCs were formed from two or three families, and two families (*Autographiviridae* and *Tectiviridae*) were split between two VGCs each (see Table 2). Core PSCs were calculated for all but three of the VGCs, because the latter comprised only one viral genome. The total number of core PSCs per VGC varied between two and one hundred and eighty-three. The VGCs with the smallest number of core PSCs mostly comprised two or three families and had low intergenomic similarities between the cluster members (see Table 2). A low number of core PSCs can be caused not only by low intergenomic similarities but also by the presence of incomplete viral genomes within the VGC. This is most likely the case of VGC#4, one of the two VGCs representing the *Autographiviridae*, which had only two core PSCs (including the signature RNA polymerase) and included many environmental viral genomes. Intuitively, high intergenomic similarities within the VGC resulted in a high number of core PSCs, especially if the phage genomes were large (for example VGC#34 representing *Molycolviridae* and VGC#31 representing *Orlanjensenviridae*).

Only for part of the core PSCs was a function annotated (see Appendix A). Most of these PSCs with known functions were involved in viral genome replication or virion morphology and morphogenesis. The major capsid protein (MCP), the portal protein, and the terminase large subunit were part of the core PSCs in 63%, 55%, and 75% of the VGCs, respectively. Within *Duplodnaviria*, the MCP at least, if not all these three VHG proteins, should be present in all viral genomes. Being more conserved than other proteins, it would be expected that the MCP would be present among the PSCs of the family-level VGCs. However, even though more conserved, these proteins can still have a high degree of sequence divergence. This can result either in the MCP assignment to different PSCs, in which case the respective PSCs would not belong to the core of a VGC, or it can hinder its recognition during the annotation step. Indeed, many of the VGCs without an MCP in the core had several core PSCs with unknown functions. Here, the annotations were performed only by comparison with the PHROG database. Generally, using the other databases (e.g., VOGDB, BLAST NR, etc) would help to retrieve more annotations.

### 3.7. A Roadmap for Using VirClust for Virus Taxonomy and Outlook

As it was recently re-affirmed in the new guidelines for virus taxonomy, the classification of viruses should reflect their evolutionary relationships [3]. This is best enabled by phylogenetic analysis of accurate multiple alignments either of complete viral genomes, when the viruses are highly related, or of their core proteins or VHGs, when the viruses are more distantly related.

Preceding phylogenetic analysis, viruses can be initially classified by constructing hierarchical clustering trees based on intergenomic distances that are not based on multiple alignments (Figure 7). For example, pairwise intergenomic distances between viral genomes can be computed by comparing their protein content, as is the case with tools like VICTOR [13], VipTree [12], and now also VirClust. The trees produced by these tools are not phylogenetic trees, because the intergenomic distances used for their calculations are not based on multiple alignments of homologous genomic regions or proteins. However, as demonstrated with VirClust on the Fam_DB dataset, they place viruses in clusters based on their relatedness. VirClust produced genomes trees that grouped related viruses at different taxonomic ranks, from order/family to genus, in a manner consistent with the current ICTV classification. In addition, VirClust enables the calculation of core proteins for the different viral clusters and the identification of realm-defining VHGs. The core proteins can be further used for the reconstruction of phylogeny inside order or family-level VGCs. The VHGs can be used for the reconstruction of more distant evolutionary relationships at the realm level, and the classification of viruses also into higher-level taxons. Finally, the hierarchical classification proposed by VirClust can be complemented and validated by phylogenetic analysis (see Figure 7), as it was successfully performed for example for the classification of novel ssDNA bacteriophages [36]. Moreover, for those viruses for which sequence divergence makes the identification of their VHGs currently impossible, phylogenetic analysis for placement at the order/family level cannot be performed and their classification will have to rely on shared genomic features, as enabled by VirClust for example.

VirClust has multiple functionalities and flexible parameters, which enables its use for various purposes, including initial taxonomic classification, calculation of core proteins and identification of VHGs. Methodologically, VirClust represents a complex pipeline, which combines novel code with the use of several stand-alone programs. The different steps can be computationally demanding, especially for larger datasets. For example, the complete analysis of one thousand nine hundred and fifty-one genomes in the dsDNA-DB dataset took about one week. The computational efficiency of VirClust can be improved further, for example by increasing the degree of code parallelization. The next version, VirClust 3, is already being developed, to enable its use on large metagenomic datasets. Feedback from the community is welcome, and suggestions for new features will be carefully taken into consideration.

## Figures and Tables

**Figure 1 viruses-15-01007-f001:**
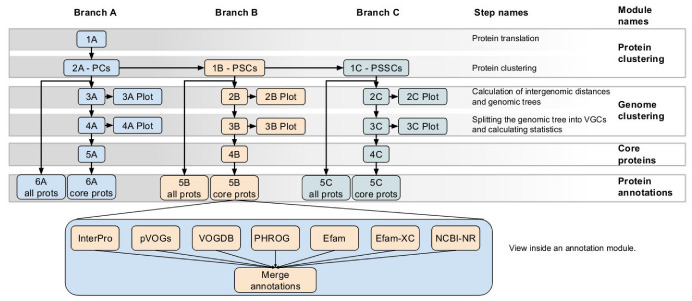
VirClust—branch and module organization. The three branches are marked in different colors. For each branch, the individual steps are labeld with a number followed by a letter (A, B or C, corresponding to each branch).

**Figure 2 viruses-15-01007-f002:**
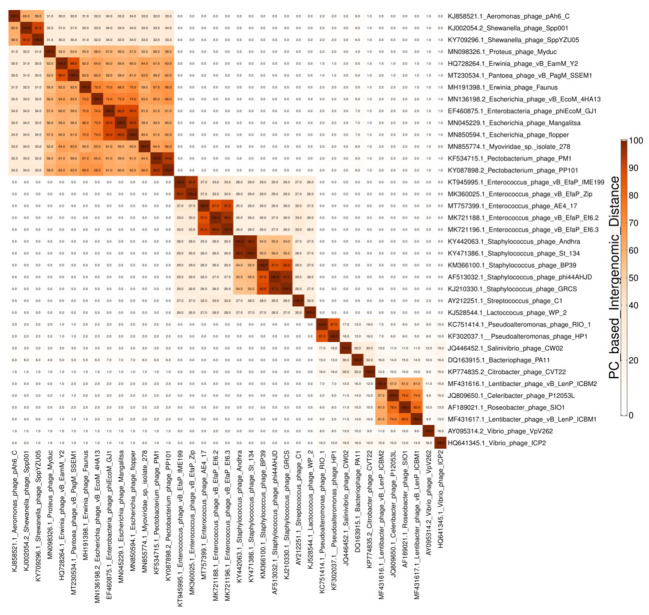
Example of a heatmap showing pairwise intergenomic similarities (%).

**Figure 3 viruses-15-01007-f003:**
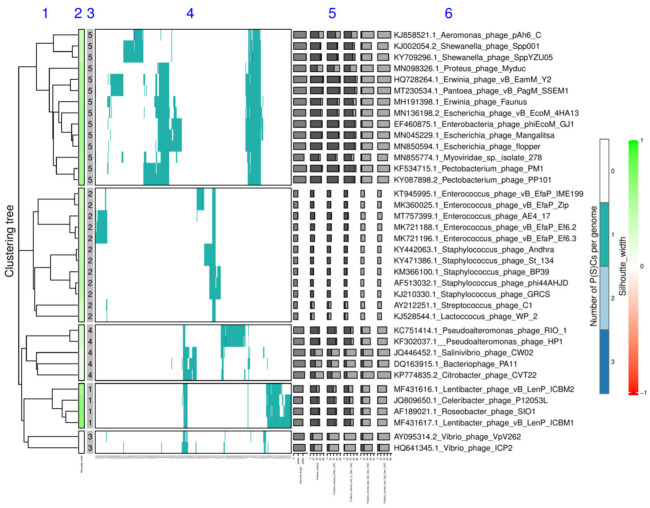
Integrated visualization of the viral clustering outputted by VirClust for the Crz_DB dataset. The genome clustering was performed based on PCs. The resulting tree was split into VGCs using a 0.9 intergenomic distance threshold. The visual components are described further. 1. Hierarchical tree calculated in step 3A, using PC-based intergenomic distances. 2. Silhouette width, color-coded in a range from −1 (red) to 1 (green). 3. VGC ID, as outputted in the genome statistic table from step 4A. 4. Heatmap representation of the PC distribution in the viral genomes. Rows are represented by individual viral genomes. Columns are represented by individual PCs. The ID of each PC can be read at the bottom of the heatmap at image magnification. Colors encode the number of each PC per genome, with white signifying the PC absence, and the other colors signifying various degrees of replication (from 1 to n, see legend). 5. Viral genome-specific statistics: genome length, the proportion of PC shared (dark grey) with any other genomes in the dataset, reported to the total PCs in the genome (light grey bar), the proportion of PC shared in its own VGC, the proportion of PCs shared only in its own VGC, the proportion of PCs shared also outside its own VGC, and the proportion of PC shared only outside own VGC. For more details about these stats, see materials and methods. 6. Virus name (here including the GenBank accession number as a suffix).

**Figure 4 viruses-15-01007-f004:**
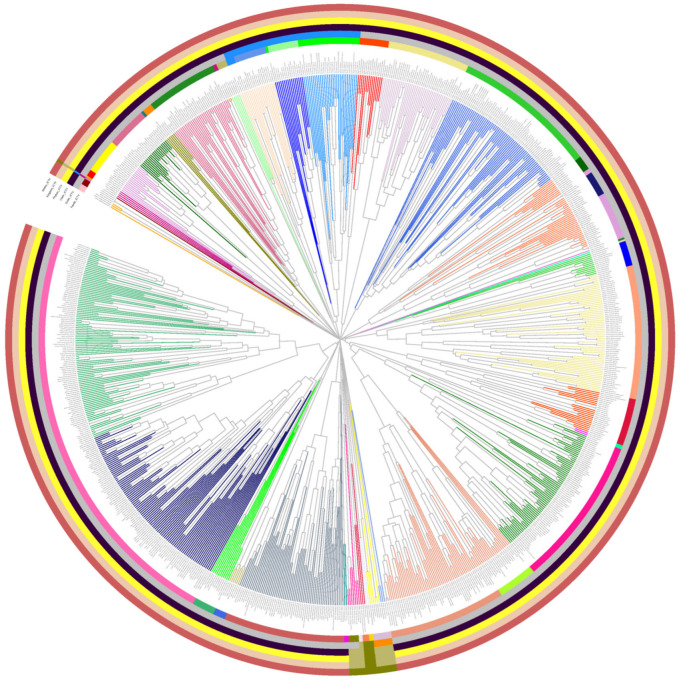
PSC-based genome tree for the Fam_DB dataset. The annotation circles encode the ICTV taxonomy for each phage genome, as follows, from inner to outer circles: family, order, class, phylum, kingdom, and realm. The colors in the circles encode the different taxons (see Appendix A for taxon names) The tree was split using a distance threshold of 0.9 and the resulting VGCs are encoded by different branch colors. The extended tree is available in Appendix A.

**Figure 5 viruses-15-01007-f005:**
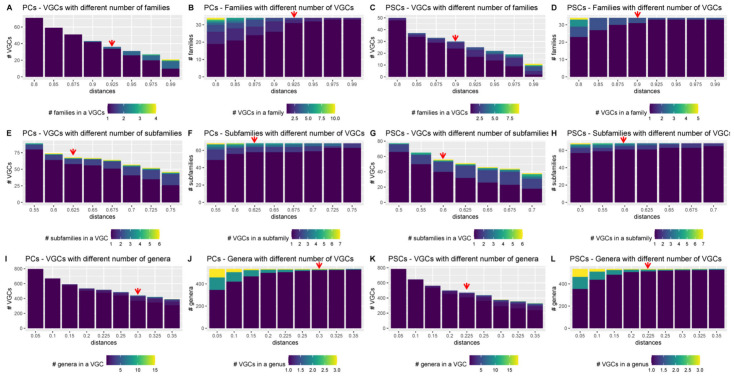
Relationship between the number of VGCs and taxons for *Duplodnaviria*, when the PC- and PSC-based genomic trees of the Fam_Db dataset were cut with different intergenomic distances. The red arrows indicate the recommended distance-hreshold to use for tree cutting, when creating VGCs. Extended figures, for all distances, are found in Appendix A.

**Figure 6 viruses-15-01007-f006:**
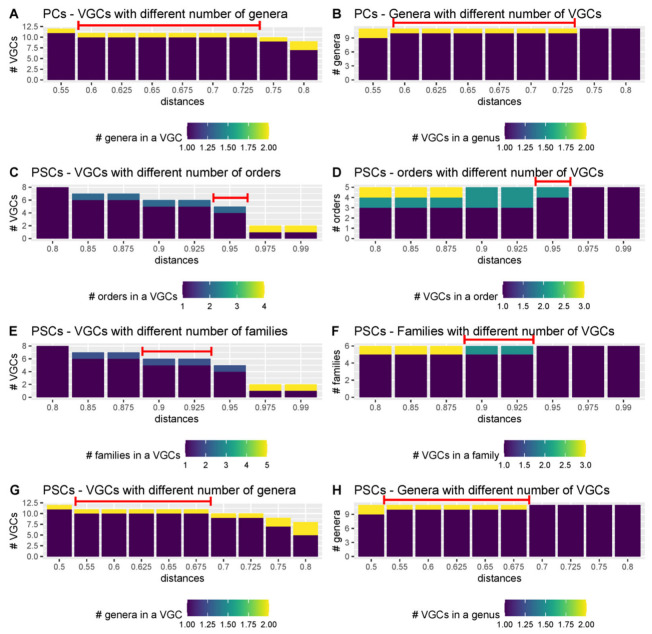
Relationship between the number of VGCs and taxons for *Varidnaviria*, when the PC- and PSC-based genomic trees of the Fam-DB dataset were cut with different intergenomic distances. The red lines indicate the recommended distance-hreshold to use for tree cutting, when creating VGCs. Extended figures, for all distances, are found in Appendix A.

**Figure 7 viruses-15-01007-f007:**
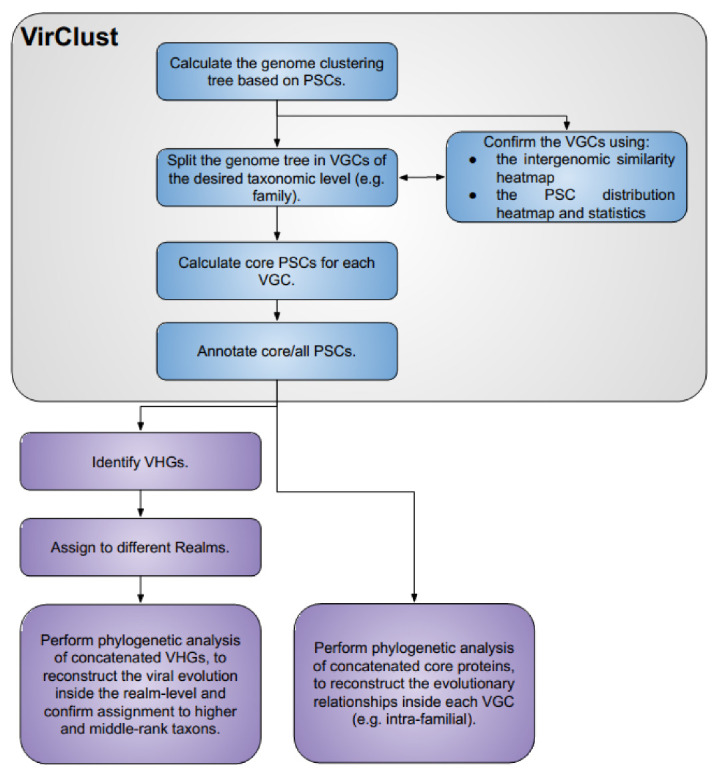
A road-map for using VirClust to enable virus taxonomy.

**Table 1 viruses-15-01007-t001:** Analysis of the Crz_DB dataset with different parameters for protein clustering. A summary of the intergenomic distances is given as the 0, 25th, 50th, 75th, and 100th percentile (pctl).

Parameters Step2A		E-Value = 10−5,	E-Value = 10−5,	E-Value = 10−5,
		Bitscore = 50,	Bitscore = 50,	Bitscore = 50,
		Coverage = 100,	Coverage = 80,	Coverage = 0,
		%id = 100	%id = 50	%id = 0
**Number of PCs**		1982	1124	805
**Intergenomic distances**				
	**0 pctl**	0.00	0.00	0.00
	**25th pctl**	1.00	0.97	0.74
	**50th pctl**	1.00	1.00	0.99
	**75th pctl**	1.00	1.00	1.00
	**100th pctl**	1.00	1.00	1.00

**Table 2 viruses-15-01007-t002:** Core PSCs for each VGCs in Fam_DB, when the clustering distance used for tree splitting was 0.9. VGCs formed from a single phage genome were excluded. The detection of four VHGs among the core PSCs is marked with a “+” in the corresponding columns. MCP = major capsid protein. TerL = terminase large subunit. Port = portal. DNApol = DNA polymerase.

VGC	Family	Number of Core PSCs		Genome Length (kbps) Range	Gene Count Range	Phage Count	Minimum Intergenomic Similarity (%)	MCP	TerL	Por	DNApol
		**Total**	**Unknown Function**								
**Duplodnaviria**											
1	Peduoviridae	7	0	28.8–40.6	37–56	59	16	+	+	+	-
3	Autographiviridae	9	0	30.8–47.7	29–65	106	17	+	+	-	+
4	Autographiviridae	2	0	10.4–47.8	15–69	99	18	-	-	-	-
5	Kyanoviridae + Ackermanviridae	32	0	144.4–252.5	178–324	76	17	+	+	+	+
6	Herelleviridae	25	3	106.1–167.5	126–294	65	14	+	+	+	+
8	Zobellviridae	10	3	38.9–49.7	55–82	11	16	+	+	-	+
9	Salamasviridae + Rountreeviridae + Guelinviridae	5	0	16.7–29	19–51	44	13	+	+	-	+
10	Drexlerviridae	23	5	44.3–51.9	62–87	40	47	+	+	-	-
11	Straboviridae	34	3	121.5–248.1	191–421	73	19	-	-	-	+
12	Steigviridae	12	4	93.6–104.6	65–119	16	14	+	+	+	-
15	Casjensviridae	17	1	54.5–64	62–88	35	26	+	+	+	-
16	Mesyanzhinovviridae	13	0	56.6–64.1	77–93	12	22	-	+	+	+
17	Demerecviridae	31	0	104.4–128.7	137–192	23	26	+	-	+	+
18	Vilmaviridae	25	0	70.2–84.4	113–151	17	22	+	+	+	+
19	Zierdtviridae	23	1	64.2–70.6	86–94	20	25	+	+	+	+
20	Schitoviridae	13	1	59.1–104	71–127	70	13	+	+	+	+
21	Chaseviridae	27	4	44.7–56.7	62–82	14	39	+	+	+	+
22	Kwiatkowskiviridae	156	92	146.4–149.9	243–274	6	69	+	+	-	+
23	Aggregaviridae + Assiduviridae	6	4	43.2–57.5	80–110	4	12	-	-	-	-
24	Pachyviridae	12	3	71.5–78.9	105–119	5	12	-	+	+	-
25	Pervagoviridae	80	56	72.6–73	84–86	2	99	-	+	+	+
27	Duneviridae + Helgolandviridae	5	3	37.6–46.6	48–63	6	12	-	-	-	-
30	Crevaviridae + Intestiviridae	23	10	83.5–98.1	77–95	20	27	+	+	+	+
31	Orlajensenviridae	24	4	17.4–17.5	24–24	3	100	+	+	+	-
33	Winoviridae	10	8	34.8–39.7	49–62	3	18	-	-	-	-
34	Molycolviridae	183	149	124.2–124.7	193–200	2	95	+	-	+	+
35	Forsetiviridae	55	37	44–47.2	66–76	2	77	+	+	+	-
36	Suoliviridae	20	10	92.6–104.1	94–172	30	19	+	-	+	+
**Varidnaviria**											
2	Matshushitaviridae	28	23	17.1–19.7	35–40	2	75	-	+	-	-
7	Corticoviridae + Autolykiviridae	3	0	10.1–10.7	16–21	7	15	+	+	-	-
13	Simuloviridae	12	11	16.4–19	29–31	3	47	-	+	-	-
14	Tectiviridae	3	0	14.4–16.6	22–30	6	11	-	+	-	+
28	Tectiviridae	8	3	17.3–18.3	30–34	3	27	-	+	-	+

## Data Availability

The stand-alone singularity image of VirClust, along with the annotation databases can be found at virclust.icbm.de. Data about the test datasets can be found as Appendix A.

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
