# Peer review of "VirClust—A Tool for Hierarchical Clustering, Core Protein Detection and Annotation of (Prokaryotic) Viruses"

_viruses, 2023, doi:10.3390/v15041007_

Round 1
Reviewer 1 Report
The tool presented here, VirClust, is extremely useful not only for the analysis of new viruses and their relationship with those previously described but also to explore the limitations of the current viral classification. Besides, it is really versatile, powerful, and flexible. I therefore, can only thank the author for having taking the time and effort to develop this tool and make it available to the community.
I only found two minor typos/formal aspects to be improved in the manuscript:
- Figure 3: Please, put the figure legend in the same page as the figure to make it easier to follow... or label the different components directly on the figure.
- Line 501: There is a typo in ViClust
Author Response
Dear reviewer
Thank you for your effort in reviewing the VirClust manuscript and for your suggestions. These have been implemented in the revised version of the manuscript.
Best regards
Cristina Moraru
Reviewer 2 Report
The manuscript describes a web server dedicated to various virus sequence-based clustering strategies. It looks like a pretty full bodied system with lots of options, and has a good user manual. There are numbers of web severs out there that do parts of what this one does, and I've generally found their performance to be pretty disappointing compared to downloading the underlying algorithms and doing it myself. This presentation talks about incorporating HMM to HMM comparison, which is a quantum leap up in sensitivity, and the manuscript does mention dealing with numbers of issues that I've noticed other attempts at this seem to naively gloss over. So I'm intrigued. The web server does exist, however the web site says that the downloadable version, which is said to be needed for "large" projects, is not yet available. So please clarify that. Are you going to make it available upon publication, or is it not finished yet?
It's a complex system. Most questions I had on first reading I was able to find the answer to on second reading, so I don't have any useful criticisms to offer. I think it's well presented and ready to be published and for some users to try it out.
The author does indicate an intention to collect user feedback and make revisions. In that vein, these are questions about interoperability that I would inevitably run into.
Can I override your gene caller with genes and start codons in my genome of interest where I think they should be, say by providing the genome in GenBank format?
Can I get an .aln of your clustalo alignments out so I can run controls on them? If I revise the alignments, can I substitute them back into the system?
For your clustergrams, you say it will compute bootstrap support values. Can I get the clustergram with support values, branch lengths, node heights, etc., out in Newick or some other standard format for reprocessing?
If I understand correctly, the downloadable version will come in an apptainer including R, hhsuite, and some databases. or is that not true? So, will I be able to search databases that didn't come in the apptainer? Will I be able to search the databases that did come in the apptainer with other software?
Author Response
Dear reviewer
Thank you for your efforts in reviewing VirClust and its manuscript.
Regarding the availability of the stand-alone version: yes, it can be downloaded from the DOWNLOAD tab of the virclust.icbm.de website. I have now updated the main page to reflect this fact as well.
Furthermore, I'm grateful for your suggestions regarding the next VirClust version. To briefly answer them:
- at the moment it is impossible to receive already predicted genes/proteins as input. However, I recognize that this would be an important addition and it is already on my to-do list for the next version.
- The aligned .fasta file can be downloaded for each PC/PSC that is aligned at steps 1B and 1C. At the moment is not possible to feed back into VirClust the improved alignments. But this is an interesting suggestion, and I can see its value. I will add it to the list of potential features for the next version.
- the clustergrams (including from the bootstrapping option) are available as .newick files
- the stand-alone version comes with several databases, which have to be downloaded separately from the DOWNLOAD tab of the virclust.icbm.de website. In this VirClust version, it is not possible to search the user's own databases. But yes, I can totally see the usefulness of such a feature. I'm not sure at the moment how to implement it, but I will think of a solution for the next version.
Thank you for all your suggestions! I have added them already to the feature list for the next VirClust version.
best regards
Cristina Moraru